# Effects of Increasing Levels of Purified Beta-1,3/1,6-Glucans on the Fecal Microbiome, Digestibility, and Immunity Variables of Healthy Adult Dogs

**DOI:** 10.3390/microorganisms12010113

**Published:** 2024-01-05

**Authors:** Pedro Henrique Marchi, Thiago Henrique Annibale Vendramini, Rafael Vessecchi Amorim Zafalon, Leonardo de Andrade Príncipe, Cinthia Gonçalves Lenz Cesar, Mariana Pamplona Perini, Thaila Cristina Putarov, Cristina Oliveira Massoco Salles Gomes, Júlio Cesar de Carvalho Balieiro, Marcio Antonio Brunetto

**Affiliations:** 1Pet Nutrology Research Center, School of Veterinary Medicine and Animal Science, University of Sao Paulo, Pirassununga 13635-000, Brazil; pedro.henrique.marchi@usp.br (P.H.M.); rafael.zafalon@usp.br (R.V.A.Z.); leoprincipe@usp.br (L.d.A.P.); cinthialenz@usp.br (C.G.L.C.); mariana.perini@usp.br (M.P.P.); balieiro@usp.br (J.C.d.C.B.);; 2Biorigin (Açucareira Quatá S.A.), Lençois Paulistas 18680-900, Brazil; thaila.putarov@biorigin.net; 3Department of Pathology, School of Veterinary Medicine and Animal Science, University of Sao Paulo, Sao Paulo 05508-270, Brazil; cmassoco@gmail.com

**Keywords:** canine, fermentation, microbiota, nutraceutical, nutrition

## Abstract

Yeast-purified beta-1,3/1,6-glucans (BG) can modulate dogs’ immune systems and microbiome, but the optimal inclusion dose remains unknown. The aim of the study was to evaluate the effects of 0.0, 0.07, 0.14, and 0.28% inclusion of BG in a dry extruded diet on the digestibility, immunity, and fecal microbiota of healthy adult dogs. Eight male and female border collies [n = 4; body condition score (BCS) = 5] and English cocker spaniels (n = 4; BCS = 5), aged 3.5 ± 0.5 years, were randomly distributed into two 4 × 4 balanced Latin squares. Fecal microbiota (using 16S rRNA sequencing, Illumina^®^), apparent digestibility coefficients (ADC) of nutrients, fecal concentrations of short-chain fatty acids (SCFA) and branched-chain fatty acids (BCFA), ammoniacal nitrogen, lactic acid, IgA and pH, lymphocyte immunophenotyping, intensity and percentage of phagocytosis and oxidative burst were determined. No differences were observed in Faith (*p* = 0.1414) and Pielou-evenness (*p* = 0.1151) between treatments, but beta diversity was different between 0.0% and 0.14% BG groups (*p* = 0.047). Moreover, the Firmicutes phylum was the most abundant in all groups and exhibited the highest relative abundance after the consumption of 0.14% BG, a finding considered beneficial for the canine microbiome. The Erysipelotrichaceae and Ruminococcaceae families, along with the *Faecalibacterium* and *Prevotella* genera, considered favorable for their involvement in butyrate production and other metabolites, showed increased abundance after the consumption of 0.14% BG. The potentially pathogenic Proteobacteria phylum displayed lower abundance after the consumption of 0.14% BG. Fecal concentrations of the evaluated compounds and pH did not differ after consumption of the BG at all percentages. Higher crude protein ADC was found after 0.14 and 0.28% BG consumption (*p* < 0.0001), but no differences were found for other nutrients. Phagocytosis, oxidative burst, and lymphocyte populations were not modulated by any of the treatments; however, 0.14% BG modulated the lymphocyte T CD4^+^:CD8^+^ ratio (*p* = 0.0368), an important marker of immune system efficiency. The inclusion of 0.14% BG resulted in the best responses and was the best dose evaluated.

## 1. Introduction

Mammals are inhabited by a high density and diversity of microorganisms, which form communities considered essential for the regular functioning of the hosts’ biological system [1]. Thus, physiological functions occur as a function of the state of dynamic and symbiotic balance between their own cells and microbial units [2]. In the canine intestinal environment, a highly complex microbiome composed of trillions of archaea, bacteria, fungi, protozoa, and viruses has been studied [3]. Nevertheless, phyla Firmicutes, Bacteroidetes, Fusobacteria, Actinobacteria, and Proteobacteria comprise more than 99% of the species identified [4].

Among the main functions of the intestinal microbiome, the ability to protect against pathogens and infections stands out. The mechanisms involved are based on the competition for metabolites and nutrients, modification of intestinal pH and oxygen, and induction of intestinal immune responses, which promotes the population reduction of potentially pathogenic bacteria [5,6].

However, this microbiome is susceptible to several factors that influence and modulate these microorganisms’ populations [7]. Among the factors stated in the scientific literature are genetics, environmental conditions, age, size, health status, pharmacological interventions, and, mainly, diet [1,8,9]. Therefore, as the search for dog diets that promote health and well-being has increased, understanding the role of nutrition in the composition of the intestinal microbiota has become a concern [9,10].

Prebiotics can have a significant impact on the composition of the intestinal microbiome, as they are used as a substrate for the fermentative metabolism of bacteria and influence their population growth or decrease [8,9]. In the dog intestinal microbiome, beneficial bacteria primarily belong to the phyla Firmicutes and Bacteroidetes and the genus *Bifidobacterium*; on the other hand, potentially pathogenic bacteria are mainly associated with the phylum Proteobacteria and the genera *Clostridium* and *Escherichia* [11].

Beta-glucans are polysaccharides composed of beta-glycosidic linkages between glucose monomers [12]. These can be extracted from the cell walls of algae, bacteria, cereals, fungi, and yeasts, each exhibiting distinct functionalities based on its polymeric structure [13,14].

*Saccharomyces cerevisiae* purified beta-1,3/1,6-glucan (BG) has been widely exploited for many years in human medicine [15], and their benefits appear to extend to various animal species [16]. BG have a marked immunostimulatory potential, but their mechanisms of action also correlate with antioxidant, antitumor, intestinal microbiota-moduLating, hypoglycemic, and hypocholesterolemic effects [16,17,18].

In dogs, studies carried out with purified beta-1,3/1,6-glucan inclusion in dog food have shown favorable effects on glucose regulation [19], lipid metabolism [20], and immune function [16,21,22]. However, among these, no studies have evaluated increasing levels of purified beta-1,3/1,6-glucan inclusion in the diet of healthy dogs.

Thus, this study aimed to evaluate the effects of 0.0, 0.07, 0.14, and 0.28% inclusion of purified BG in a dry extruded diet on digestible, immunity, and fecal microbiota variables of healthy adult dogs, with the hypothesis that the minimum inclusion of BG in the diet results in beneficial effects on these variables.

## 2. Materials and Methods

This study was carried out in agreement with the Ethical Principles in Animal Research established by the Ethic Committee on Animal Use of the School of Veterinary Medicine and Animal Science at the University of Sao Paulo (CEUA/FMVZ). The study was approved under protocol number 2866090223.

### 2.1. Location and Animals

The experiment was conducted at the Pet Nutrology Research Center (CEPEN Pet) of the Animal Nutrition and Production Department of the School of Veterinary Medicine and Animal Sciences—University of Sao Paulo (FMVZ/USP), in the city of Pirassununga, Sao Paulo, Brazil. Eight healthy male and female dogs (four border collies and four English cocker spaniels), neutered, with a mean age of 3.5 ± 0.5 years, an ideal body condition score (BCS) [23], and an average BW of 17.55 kg ± 5.47 were included.

The animals had their health previously evaluated through a complete physical examination, nutritional anamnesis, complete blood count, and biochemical profile tests [albumin, glucose, total protein, urea, creatinine, alkaline phosphatase, cholesterol, triglycerides, aspartate aminotransferase (AST), and alanine aminotransferase (ALT)] in order to assess the health status of the animals.

The dogs were housed in collective kennels with 3.42 m^2^ of covered area and 7.21 m^2^ of uncovered area, concrete floors, and tiled walls. During the sample collection periods, the animals remained housed in individual kennels. All animals had access to fresh water ad libitum.

### 2.2. Diets and Experimental Design

The animals were randomly distributed into two balanced 4 × 4 Latin squares, containing four experimental periods and four experimental dry extruded diets without beta-glucan inclusion (0.0% BG), with 0.07% beta-glucan inclusion (0.07% BG), with 0.14% beta-glucan inclusion (0.14% BG), and 0.28% beta-glucan inclusion (0.28% BG). These inclusion levels encompass the range of beta-1,3/1,6-glucan doses employed in previous studies [16,19,20,21,22]. The diets were prepared in the feed factory of the School of Veterinary Medicine and Animal Science of Sao Paulo State University (FMVZ/UNESP), Botucatu, Sao Paulo, Brazil (Table 1). Energy intake for each animal was estimated at 95 kcal × body weight^0.75^ a day [24]. The purified beta-glucan (MacroGard^®^, Açucareira Quatá S.A-Biorigin, Lençóis Paulista, Brazil) was extracted from the *Saccharomyces cerevisiae* cell wall through a biotechnological process, where the layer of mannanoligosaccharides was carefully removed to keep the functionality of the beta-1,3/1,6-glucan structure. The BG were added to dog food prior to the extrusion process.

The study lasted a total of 140 days, and each period took 35 days: the first 28 days were intended for diet acclimation, and the following 7 days were destined for sample collection. From days 29 to 33, total fecal samples were collected to measure ADC. On day 34, fresh feces were aseptically placed into 2 mL vials and immediately frozen at −80 °C for later determination of fecal microbiota, fermentative products, and IgA. Finally, on day 35, 10 mL blood samples were collected from the jugular venipuncture for immunological variable analyses and measured on the same day (Figure 1).

### 2.3. Fecal Microbiota

Fresh feces were collected and stored in 2 mL vials at −80 °C. The samples did not have contact with any other surface besides sterile gloves. Determination of the fecal bacterial population was performed using Illumina^®^ Sequencing technology.

Total DNA extraction was performed using the Quick-DNA Fungal/Bacterial DNA MiniPrepTM kit (Zymo Research, Irvine, CA, USA), according to the manufacturer’s protocol. PCR reactions were performed in a final volume of 20 μL, including 10 μL of GoTaq^®^ Green PCR Master Mix (Promega, Madison, WI, USA), 1 μL of 5 μM forward oligonucleotide, 1 μL of 5 μM reverse oligonucleotide, 1 μL of genomic DNA, and sterile ultrapure water, sufficient for a total volume of 20 μL. Two different primers were used for amplification, and the reactions were conducted in a Veriti™ Thermal Cycler (Applied Biosystems, Foster City, CA, USA).

The amplification program consisted of an initial denaturation at 95 °C for 3 min, followed by 27 cycles of denaturation at 95 °C for 30 s, annealing at 55 °C for 30 s, extension at 72 °C for 30 s, and a final extension at 72 °C for 5 min. Amplification products were confirmed using electrophoresis on a 2.0% agarose gel stained with UniSafe Dye 0.03% (*v*/*v*) ~600 bp (amplicon size).

Indexing was performed according to the Nextera XT Index kit protocol (Illumina). The indexing program consisted of incubation at 72 °C for 3 min, initial denaturation at 95 °C for 30 s, followed by 12 cycles of 95 °C for 10 s, 55 °C for 30 s, and 72 °C for 30 s, and a final extension at 72 °C for 5 min. Amplification reactions were performed in a Veriti™ Thermal Cycler (Applied Biosystems, Foster City, CA, USA).

The generated libraries were subjected to purification steps using Agencourt AMPure XP magnetic beads (Beckman Coulter, Brea, CA, USA) to remove very small fragments of the total population of molecules and residual primers. Quantification was then performed using the KAPA PCR-based method for real-time library quantification (Library Quantification Kit—Illumina/Universal) on the QuantStudio 3 instrument (Applied Biosystems, Foster City, CA, USA), all according to the manufacturer’s protocol. An equimolar DNA pool was generated by normalizing all samples to 4 nM for sequencing, which was performed using the Illumina MiSeq next-generation sequencing system (Illumina^®^ Sequencing) and the MiSeq Reagent V2 Nano 500 cycles kit—PE 2 × 250 bp. The sequencing cartridge was loaded with 20% Phix (PhiX Control v3—Illumina, San Diego, CA, USA).

These determinations were carried out at the BPI Biotecnologia biotechnology research laboratory located in Botucatu, SP, Brazil.

### 2.4. Apparent Digestibility Coefficients of Nutrients

Dietary apparent digestibility coefficients (ADC) of nutrients were determined using the total fecal collection method according to AAFCO [25]. In summary, food consumption was recorded daily, and total feces were collected for five days. Stools were weighed immediately after collection, placed in individual plastic bags, and stored in a freezer (−15 °C) for further analysis. At the end of the collection period, total feces were weighed and dried in a forced ventilation oven (320-SE, FANEM, Sao Paulo, Brazil) at 55 °C for at least 72 h until moisture content decreased below 10%. The pre-dried stools and diets were then ground in a knife mill (MOD 340, ART LAB, Sao Paulo, SP, Brazil) with a 1 mm sieve and stored in plastic jars at room temperature until laboratory analyses.

The dry matter, crude protein, ethereal extract in acid hydrolysis (fat), crude fiber, and ash content of the feces and food were determined according to the method described by the AOAC [26]. In addition, the calcium and phosphorus contents of foods were also determined [26]. The nitrogen-free extractives were calculated by the difference between dry matter and the sum of crude, fat, crude fiber, and ashes. All bromatological analyses were carried out at the Multiuser Laboratory of Animal Nutrition and Bromatology of the Department of Animal Nutrition and Production of FMVZ/USP, located in Pirassununga.

Based on the results obtained in the laboratory, the apparent digestibility coefficients of dry matter, organic matter, crude protein, fat, crude fiber, and non-nitrogen extractives of feeds were calculated. These calculations were performed using the following equation:ADC (%) = [nutrient intake (g) − nutrient output (g)]/[nutrient intake (g)] × 100

### 2.5. Fermentative Products

#### 2.5.1. Fecal Ammoniacal Nitrogen, Short-Chain Fatty Acids, and Branched-Chain Fatty Acids

Fecal ammoniacal nitrogen, short-chain fatty acids (SCFA), and branched-chain fatty acids (BCFA) were determined on fresh, homogenized fecal samples collected within 15 min after defecation. Immediately upon collection, three grams of duplicated fecal samples for each parameter measured were mixed with 9 mL 16% formic acid. The mixture was kept in a refrigerator for seven days and stirred daily. Subsequently, the mixture was centrifuged at 5000 rpm for 15 min at 15 °C three times, discarding the precipitate. The supernatants were extracted, identified, and stored at −15 °C.

For fecal ammonia nitrogen quantification, the extracts were thawed at room temperature, and 2 mL aliquots were diluted in 13 mL distilled water and then processed in a nitrogen distiller, according to Vieira [27]. These analyses were performed at the Multiuser Laboratory of Animal Nutrition and Bromatology of the Department of Animal Nutrition and Production of FMVZ/USP, located in Pirassununga.

For SCFA and BCFA determination, all samples were thawed and centrifuged at 14,000 rpm (Rotanta 460 Robotic, Hettich, Tuttlingen, Germany) for 15 min. Fecal concentrations were analyzed using gas chromatography (SHIMADZU, model GC–2014, Kyoto, Japan), according to Erwin et al. [28]. The analysis was performed using a 30 m × 0.53 mm glass column (Stabilwax1, Restek, Bellefonte, PA, USA) at 145 °C and nitrogen as carrier gas at a flow rate of 8.01 mL/min. The working temperatures were: injection, 250 °C; column, 145 °C (at a speed of 20 °C/min); and flame ionization detector, 250 °C. These analyses were performed at the Faculty of Animal Science and Food Engineering (FZEA/USP) Ruminal Fermentability Laboratory, Pirassununga, SP, Brazil.

#### 2.5.2. Lactic Acid Analysis

Each sample was homogenized and mixed with 2 mL of distilled water (1:2 *w*/*v*). These mixtures were kept for a period of three days in a refrigerator (5 °C) and were homogenized daily. After this period, the samples were centrifuged for 5 min at 2852 rpm (Fanem 206-R Centrifuge Excelsa Baby II, Sao Paulo, Brazil) and the supernatant was used. In a test tube, 1 mL of the supernatant and 6 mL of sulfuric acid were placed, which, after being stirred, were placed in boiling water at 80 °C for three minutes. After the tubes had cooled down, 0.1 mL of a solution containing 1.5 g of p-hydroxybiphenyl and 100 mL of dimethylformamide was added, and the tubes were placed again in boiling water at 80 °C for 90 s. Lactic acid was measured according to the methodology described by Pryce [29], using the spectrophotometry method at 565 nm (500 to 570 nm), using reagent blank in order to calibrate the spectrophotometer (Nova 2000 UV Spectrophotometer, Piracicaba, Brazil). Samples were quantified by comparing them with a 0.08% lactic acid standard. These analyses were carried out in triplicate at the Multiuser Laboratory of Animal Nutrition and Bromatology of the Department of Animal Nutrition and Production of FMVZ/USP, located in Pirassununga, SP, Brazil.

#### 2.5.3. Fecal pH Measurement

One gram of fresh feces was weighed and diluted in 9 mL of distilled water according to the adapted method from Walter et al. [30]. The pH evaluation was carried out using a digital benchtop pH meter with an autonomous electrode (Starter 3100, pH Bench, Ohaus, Sao Paulo, SP, Brazil). After preparation, the electrode was directly introduced into a 9:1 solution of distilled water and feces and computed pH. These determinations were carried out at the Pet Nutrology Research Center (CEPEN Pet) of FMVZ/USP, Sao Paulo, SP, Brazil.

### 2.6. Immunological Variables

#### 2.6.1. Lymphocyte Immunophenotyping

Blood samples were diluted with PBS at a 1:1 ratio. In a 15 mL sterile conical centrifuge tube, 2 mL of the Ficoll–Parque density gradient and 1 mL of the diluted blood were inserted. The constituent was centrifuged for 20 min at 400× *g* and 20 °C to separate interface mononuclear cells, and then the material was placed in a cryogenic tube and stored in a −80 °C freezer.

The helper T lymphocytes (CD4^+^), cytotoxic T lymphocytes (CD8^+^), and the CD4^+^:CD8^+^ ratio were evaluated. Mononuclear cells (2 × 10^5^ cells/mL) were incubated in microtubes (1.5 mL) with CD3+ (1:100) and CD4 (1:10) antibodies (Biolegend, San Diego, CA, USA), diluted in 100 µL of buffer for cytometry (PBS containing 0.5% bovine serum albumin and 0.02% sodium azide). Isotype antibodies were included in the assay to define the negative region (background). Cells were incubated for 20 min at 4 °C, protected from light. After the end of the incubation period, the samples were washed twice with buffer for cytometry in a volume of 1000 µL per microtube. Finally, the cells were resuspended in 500 µL of PBS. Cells with low size (FSC) and low complexity (SSC) were selected as lymphocyte populations, according to the delimited gate. From this selection, the different populations of lymphocytes were determined. The acquisition and analysis of 10,000 cells were performed using the FACS Calibur model flow cytometer (Becton Dickinson, San Jose, CA, USA). These determinations were carried out at the Comparative Immuno-oncology Laboratory at FMVZ/USP, located in Sao Paulo, SP, Brazil.

#### 2.6.2. Phagocytosis and Oxidative Burst Test

Blood leukocytes (lymphocytes, neutrophils, and monocytes) were incubated with a fluorescent reagent that indicates the production of reactive oxygen species at baseline and after the phagocytosis of *Staphylococcus aureus* bacteria. Cells were incubated with DCFH-DA reagents in PBS (phosphate buffered saline) and DCFH and *S. aureus* labeled with a fluorochrome (propidium iodide) and then kept at 37 °C for 20 min. After this period, the red blood cells were broken up using a lysis solution and washed with PBS until a clear-looking sample was obtained, which was read on a FACS Calibur model flow cytometer (Becton Dickinson, San Jose, CA, USA). These analyses were carried out at the Comparative Immuno-oncology Laboratory at FMVZ/USP, located in Sao Paulo, SP, Brazil.

#### 2.6.3. Fecal Immunoglobulin A Analysis

After thawing, IgA was extracted using a saline solution, according to the method described by Peters et al. [31]. One gram of feces was weighed and added to 10 mL of extraction buffer [0.01 M PBS, pH 7.4, 0.5% Tween 80 (Sigma-Aldrich, Poole, Dorset, UK) and 0.05% sodium azide] and homogenized. The suspension was centrifuged at 1500× *g* for 20 min at 5 °C.

From the supernatant, 2 mL were transferred to a 5 mL conical tube containing 20 μL of a protease inhibitor cocktail (Sigma-Aldrich, Sao Paulo, Brazil), homogenized, and centrifuged at 15,000× *g* for 15 min at 5 °C. The supernatant was transferred to an Eppendorf tube and stored at −20 °C. IgA quantification was performed using an ELISA kit for canine IgA (Bethyl Laboratories, Montgomery, TX, USA) according to the manufacturer’s recommendations. The reading was performed in an ELISA Microplate Reader (Microplate Reader MRX TC Plus, Dynex Technologies, Chantilly, VA, USA) equipped with a 450 nm filter. These determinations were carried out at the LEAC laboratory, located in Sao Paulo, SP, Brazil.

### 2.7. Statistical Analyses

The design used was a contemporary double Latin square with four treatments. For the variables of apparent digestibility of nutrients, fermentative products, and immunological parameters, a general mixed linear model was applied, considering the fixed effect of treatment (0.00, 0.07, 0.14, and 0.28% BG inclusion), as well as the random effects of the animal within square, period, and residue. To evaluate the observed abundances for each phylum, family, and genus, a generalized mixed linear model was adopted, considering the binomial distribution of abundances and using the logit link function. The statistical model included the fixed effect of treatment and the random effects of the animal within the square, period, and residue. In case of significant results in the analysis of variance type 3, the Tukey test was used as a multiple-comparison procedure. All analyses were performed using the PROC MIXED and PROC GLIMMIX procedures, respectively, of the Statistical Analysis System program, version 9.4 (SAS Institute Inc., Cary, NC, USA).

## 3. Results

### 3.1. Animals and Diet Compositions

Throughout the experiment, all animals remained healthy, and no episodes of emesis or diarrhea were reported. The follow-up of food intake during the experimental periods confirmed the complete ingestion of meals and the maintenance of body weight (BW) and BCS [23] of the dogs. Therefore, it was not necessary to make dietary adjustments or exclude any animal from this study.

In general, the chemical analysis of the four experimental diets showed similar composition values (Table 2). However, it was possible to highlight an increase of approximately 3.0% in the percentage of crude protein between 0.0% and 0.07% compared to 0.14% and 0.28% treatments.

The average daily consumption of the diets was 221.38 g ± 53.88 during the entire experimental period of the study. Therefore, considering that the product used has 60% BG, the intake of this prebiotic in each treatment was 0 mg/kg after 0.0% BG consumption, 5.30 mg/kg after 0.07% BG consumption, 10.60 mg/kg after 0.14% BG consumption, and 21.20 mg/kg after 0.28% BG consumption.

### 3.2. Fecal Microbiota

Among the results obtained by sequencing the animals’ fecal microbiota and bioinformatics analysis, no differences were found in alpha diversity when analyzing Faith’s phylogenetic diversity (*p* = 0.1414; Figure 2) and Pielou-evenness indexes (*p* = 0.1151; Figure 3).

The determination of beta diversity, evaluated using the 3D principal coordinate analysis performed using the QIIME program version 2022.9 (Figure 4), showed a grouping of experimental treatments between the axes, which assumes a brief taxonomic variation between the four groups. However, it was possible to observe taxonomic variability between the intestinal microbiome after 0.0 and 0.14% BG consumption (*p* = 0.047).

Based on sequencing analysis, 6 phyla, 32 families, and 82 distinct genera were identified from samples from the four experimental groups. Due to the low number of observations of certain phyla, families, and genera in the samples, these were excluded from the statistical analysis since the low representativeness made it impossible to compare treatments.

Of the four groups, Firmicutes phylum had the highest mean relative abundances (34.18–58.54%), and Proteobacteria (2.21–1.42%) and Actinobacteria (0.90–0.15%) had the smallest. It was possible to observe that, after the consumption of 0.0%, 0.07%, and 0.28% diets, the second and third highest relative abundances were represented by the phyla Fusobacteria (32.32–23.92%) and Bacteroidetes (26.38–19.75%), respectively. However, the opposite scenario was observed in dogs that consumed the 0.14% BG, which had a greater abundance of Bacteroidetes (21.30%) compared to Fusobacteria (14.09%; Figure 5).

Different behaviors were observed regarding the relative abundance of the five phyla analyzed depending on the BG inclusion level in the diet (Table 3). Actinobacteria was reduced after 0.07% BG consumption and increased after 0.28%; however, there was no difference between treatments 0.00 and 0.14% (*p* < 0.0001). For Bacteroidetes, a reduction was observed after the consumption of the three BG treatments compared to the 0.0% group (*p* < 0.0001). The group that shows the greatest reduction in the relative abundance of this phylum was 0.07% BG when compared with 0.14 and 0.28%, which did not differ between themselves. The reduction after BG consumption was also observed in the phyla Fusobacteria and Proteobacteria, whose relative abundances were the lowest after consumption of 0.14%, 0.07%, and 0.28% BG groups did not differ from each other but presented lower values when compared to the 0.0% group (*p* < 0.0001). In contrast to the previous, the Firmicutes phylum was positively modulated after the consumption of BG. When compared to the 0.0% group, the treatment that better increased the relative abundance of these bacteria was 0.14%, followed by 0.07 and 0.28%, which did not differ from each other (*p* < 0.0001).

Regarding the total number of characterized families, 16 were analyzed, and another 16 were excluded from the statistical analysis. The three families with the highest average relative abundance were Bacteroidaceae (16.17–11.76%), Fusobacteriaceae (32.32–14.09%), and Lachnospiraceae (22.89–15.18%). The population ranking of each at 0.0% inclusion was, in descending order, Fusobacteriaceae, Bacteroidaceae, and Lachnospiraceae. Treatments 0.07% and 0.28% showed a higher abundance of Lachnospiraceae compared to Bacteroidaceae. Finally, the 0.14% BG treatment was different from the previous ones, as it had the highest relative abundance for Lachnospiraceae, followed by Fusobacteriaceae and Bacteroidaceae (Figure 6).

Of the total number of families analyzed, a difference was observed for 15 of them, according to the dosage of BG intake (Table 4). A positive modulation was observed in the families Coriobacteriaceae (*p* = 0.0111), Erysipelatoclostridiaceae (*p* < 0.0001), Lachnospiraceae (*p* < 0.0001), Peptostreptococcaceae (*p* = 0.0003), Ruminococcaceae (*p* < 0.0001), Selenomonadaceae (*p* < 0.0001), and Succinivibrionaceae (*p* < 0.0001). Furthermore, negative modulation was observed for Acidaminococcaceae (*p* < 0.0001), Bacteroidaceae (*p* < 0.0001), Fusobacteriaceae (*p* < 0.0001), Prevotellaceae (*p* < 0.0001), Sutterellaceae (*p* < 0.0001), and Tannerellaceae (*p* = 0.0438) after consuming different levels of BG. Specifically, the family Erysipelotrichaceae (*p* < 0.0001) showed increases and decreases in their relative abundances compared to 0.0% BG. Furthermore, the Clostridiaceae family (*p* = 0.0011) was modulated only in treatments of 0.07 and 0.14%. No difference was observed in the relative abundance of Peptococcaceae after consumption of the four experimental treatments (*p* = 0.3061).

The lowest taxonomic classification obtained through sequencing was the bacteria genus. Of the 82 genera characterized, 54 were excluded due to low representation, and the remaining 28 were analyzed (Figure 7). Significant variations were found in 25 average relative abundances (Table 5). In general, the genera whose relative abundances were positively modulated by at least one of the treatments containing BG were *Anaerobiospirillum* (*p* < 0.0001), *Blautia* (*p* < 0.0001), *Collinsella* (*p* = 0.0111), *Erysipelatoclostridium* (*p* = 0.0008), *Faecalibacterium* (*p* < 0.0001), *Lachnospiraceae NC2004* (*p* < 0.0001), and *Megamonas* (*p* < 0.0001). The following genera were observed to be negatively modulated after BG consumption: *Alloprevotella* (*p* < 0.0001), *Bacteroides* (*p* < 0.0001), *Fusobacterium* (*p* < 0.0001), *Lachnospira* (*p* = 0.0002), *Lachnospiraceae UCG*-0004 (*p* < 0.0001), *Parabacteroides* (*p* = 0.0191), *Parasuterella* (*p* = 0.0002), *Phascolarctobacterium* (*p* < 0.0001), *Prevotellaceae Ga6A1* (*p* < 0.0001), and *Suterella* (*p* = 0.0002). The genera *Allobaculum* (*p* < 0.0001), *Prevotella* (*p* < 0.0001), *Turicibacter* (*p* < 0.0001), and *Ruminococcus* (*p* < 0.0001) exhibited both increases and decreases in their relative abundances when compared with the 0.0% BG group. Furthermore, the genus *Clostridium sensu stricto 1* differed only between the 0.14% (1.09% ± 0.61) and 0.07% (0.71 ± 0.40%; *p* = 0.0012) groups. Similarly, *Fournierella* showed a difference in its modulation only between treatments with BG inclusion. In this case, treatment 0.28% (0.55 ± 0.14%) differed only when compared to treatment 0.07% (0.24% ± 0.06; *p* = 0.0075). Bacteria of the genus *Peptoclostridium* showed greater relative abundance in the 0.07% group and were negatively modulated after the consumption of 0.14 and 0.28% BG (*p* < 0.0001). Finally, no difference was observed in the relative abundances of *Holdemanella* (*p* = 0.2969), *Peptococcus* (*p* = 0.3061), and *Romboutsia* (*p* = 0.2994) after the consumption of the four experimental treatments.

### 3.3. Apparent Digestibility Coefficients of Nutrients

A difference was observed between averages of the ADC of crude protein after 0.14% and 0.28% BG consumption when compared to 0.00 and 0.07% (*p* < 0.0001). No differences were observed between dry matter, organic matter, fat, ash, crude fiber, and nitrogen-free extract (Table 6).

### 3.4. Fermentative Products

No differences were observed between the average concentrations obtained in the four BG groups for ammoniacal nitrogen, lactic acid, fatty acids, and pH in the dog feces samples (Table 7).

### 3.5. Immunologic Variables

A difference was observed between the mean of the T CD4^+^:CD8^+^ ratio of the 0.14% BG group compared to the 0.0% BG group (*p* = 0.0368; Figure 8). Regarding the percentage and intensity of phagocytosis, oxidative burst, individual populational percentage of T CD4^+^ and T CD8^+^ lymphocytes, and fecal immunoglobulin A (IgA) concentration, no differences were observed (Table 8).

## 4. Discussion

*Saccharomyces cerevisiae*-purified beta-1,3/1,6-glucan supplementation in dog food is widespread in the pet industry and provides a series of scientific evidence that corroborates the benefits of this nutraceutical [13,16,33,34]. However, this is the first study that sought to evaluate the effects of increasing levels of purified beta-1,3/1,6-glucan inclusion in the diet of healthy adult dogs on fecal microbiota, digestive, and immunological variables. Thus, as a first contribution, it can be stated that the inclusion of purified beta-1,3/1,6-glucan in dry extruded food for dogs proved to be safe since no clinical changes or impacts on the quality of the animals’ feces were observed during the experiment, as observed in other studies [13,16,17,33,34].

The yeast cell wall has been recognized as a prebiotic for several years, and its benefits on intestinal health are well established in the literature [35,36,37]. *Saccharomyces cerevisiae* purified beta-glucan is carefully extracted from the yeast cell wall to preserve and segregate the 1,3/1,6 functionality.

The data obtained through the metagenomic sequencing of the microbiome allow us to determine the differences between the experimental groups under the diversity metric within each sample, called alpha, and the diversity evaluated simultaneously between the samples, called beta [38]. Alpha metrics represent the structure of the microbial community, which includes richness, determined by the number of taxonomic groups, and uniformity, which corresponds to the abundance distribution of groups [39]. In this study, Faith [40] was used for the phylogenetic diversity index, which is quite common and recurrent in microbiome studies.

The phylogenetic assessment of beta diversity allows observing how different the microbiome of an experimental group is from another by measuring changes in microbial composition or structure through the abundance, presence, or absence of gene sequences [38,41]. The analysis used in the study was based on the permutational multivariate ANOVA test, also referred to as PERMANOVA [42]. Therefore, the difference observed for this metric demarcates taxonomic variability of the fecal microbiota between the 0.14% BG group and the control animals in the 0.0% BG group.

Regarding population variations in different taxonomic categories, it was possible to observe many changes in the relative abundances of phyla, families, and genera between treatments. The Firmicutes phylum had the highest relative abundance in all four treatments, a finding also identified by other authors [8,37,43]. Furthermore, the highest modulation in the estimated average of relative abundances was observed in this phylum (>24%) after 0.14% BG consumption. In general, the Firmicutes phylum is considered beneficial for the intestine despite being very heterogeneous and comprising several phylogenetic groups [44,45]. This phylum comprises gram-positive bacteria that are predominantly from the genera *Bacillus*, *Clostridium*, *Enterococcus*, *Lactobacillus*, and *Ruminococcus* [46]. Despite the usual association that *Clostridium* bacteria are pathogenic, these microorganisms perform complex beneficial functions in the intestine [47]. For example, bacteria from the Clostridiaceae family are able to degrade fiber and produce SCFA in the intestine [8] and have recently been correlated with protein metabolism and improved stool quality in dogs [48]. However, the representativeness of the Clostridiaceae family was low and only differed between treatments with 0.14% BG, which presented a higher relative mean than the 0.07% BG group.

The Erysipelotrichaceae family was one of those responsible for enhancing the increase in the relative abundance of Firmicutes, especially in the 0.14% BG treatment, in which it presented the highest mean. In humans, the Erysipelotrichaceae family has already been correlated with several diseases, including inflammatory and metabolic intestinal disorders [49]. However, healthy dogs exhibited a greater abundance of these bacteria when compared to dogs suffering from acute hemorrhagic diarrhea [50]. Furthermore, dogs diagnosed with inflammatory bowel disease showed a reduction in Erysipelotrichaceae populations [51,52]. In this way, the evidence suggests that this family is beneficial for canine intestinal health. Moreover, it seems to participate in the digestion of carbohydrates and fibers, leading to the production of SCFA [48].

Furthermore, the Erysipelotrichaceae family is diverse in the microbiome and comprises genera also considered beneficial for dogs, such as *Allobaculum* and *Catenibacterium* [53]. The relative abundance of the first genus behaved in synergy with what was observed in its family and, therefore, presented the highest relative abundance in the 0.14% BG group. In turn, the *Catenibacterium* genus was identified in the samples with low representativeness; thus, it was removed from the statistical analysis.

Also contributing to the increase in Firmicutes, the Ruminococcaceae family and its respective genus *Faecalibacterium* exhibited a rise in relative abundance in the fecal microbiota of dogs following the 0.14% BG consumption. Treatments 0.07 and 0.28% BG showed intermediate estimated means, higher than those observed in dogs fed 0.0% BG. Similar to the previous bacteria, both are considered favorable for the intestinal environment since they are butyrate producers [48]. Furthermore, decreased *Faecalibacterium* abundance has previously been associated with acute diarrhea in both dogs and cats [54,55].

In a study by Suchodolski et al. [50] comparing the fecal microbiota of healthy dogs and dogs with enteropathies, distinct patterns were observed. Dogs with acute diarrhea showed significant reductions in *Turicibacter*, *Ruminococcus*, *Faecalibacterium*, and *Blautia* and a marked increase in *Sutterella* and *Clostridium perfringens*. In the present study, 0.28% BG had the highest relative abundance for *Turicibacter*, while treatment 0.14% did not differ from the 0.0% control, and treatment 0.07% had the lowest average. For *Blautia*, treatments 0.07 and 0.28% BG had the highest estimated means of abundance, while 0.14% did not differ from 0.0%. Regarding *Ruminococcus*, the highest average abundances were observed in treatments of 0.07%, followed by 0.14%, 0.0%, and 0.28%. Lastly, all treatments with BG inclusion showed a reduction in *Sutterella* compared to the control animals. These findings imply balanced benefits between treatments and indicate that the dogs supplemented with BG exhibited improved intestinal health, as per Suchodolski et al. [50] criteria distinguishing healthy and diseased dogs.

Among the treatments, the second and third most present phyla were distinct due to a marked reduction in the relative abundance of Fusobacteria in the 0.14% BG group. Overall, the phylum was modulated due to observed decreases in the abundance of the Fusobacteriaceae genus *Fusobacterium*. In humans, these microorganisms have been linked to the development of conditions such as inflammatory bowel disease [56,57]. However, evidence is mixed in healthy and diseased dogs. Vazquez-Baeza et al. [53] suggested that dysbiosis parameters in dogs with inflammatory bowel disease are divergent from those proposed for humans. In this context, the *Fusobacterium* genus was considered beneficial for the intestinal microbiota of dogs in this and other studies [52,53,58].

On the other hand, the reduction in *Fusobacterium* after the consumption of *Saccharomyces cerevisiae* BG and some prebiotic ingredients has already been demonstrated by other studies [37,59,60]. In a study by Macedo et al. [43], obese dogs exhibited a higher relative abundance of *Fusobacterium* when compared to lean dogs. Moreover, weight loss in obese animals resulted in a decrease in abundance at an intermediate level between groups. Lastly, this genus was negatively correlated with the production and concentration of intestinal butyric acid [48]. These varying pieces of evidence create a challenge in comprehending the precise biological functions of *Fusobacterium*. Consequently, it remains inconclusive whether the observed reductions are positive or negative for the canine intestinal microbiota.

Bacteroidetes constituted the third most abundant phylum in the fecal microbiota of dogs under treatments 0.0, 0.07, and 0.28% BG and the second most abundant after 0.14% BG consumption. It is noteworthy that all treatments with BG inclusion showed a reduction in the relative abundance of Bacteroidetes. Prior studies found no difference in the relative abundance of Bacteroidetes in the fecal microbiota of dogs supplemented with yeast-based products and certain prebiotics [7,37,58,59,60,61]. In studies conducted by Panasevich et al. [7] and Santos et al. [37], although no differences were noted for Bacteroidetes, reductions were observed in the abundance of the *Prevotella* genus in the fecal microbiota of dogs supplemented with potato fiber and cultured yeast, respectively. The *Prevotella* genus is known to be involved in the fermentative metabolism of carbohydrates, which have SCFA as an end product [44].

Despite the reduction in Bacteroidetes, the 0.14% and 0.28% BG treatments exhibited increases in the relative abundance of *Prevotella*, suggesting a beneficial effect of the supplement on the canine intestinal microbial population. Lastly, a reduction in the Bacteroides genus was observed in all BG treatments. Bacteroides have conflicting evidence, as dogs with chronic diarrhea have shown both increases [62] and decreases [63,64] in relative abundance. This discrepancy complicates discussions about its potential benefits in healthy dogs.

The two smallest phyla found were Actinobacteria and Proteobacteria, a finding supported by previous studies [8,37,43,59]. Concerning Actinobacteria, a moderate decrease was noted after the 0.07% BG treatment, with a lower average abundance observed after the 0.28% BG diet. Notably, the 0.14% BG diet did not significantly differ from the 0.0% group. The main representative genus of Actinobacteria is *Bifidobacterium*, which were identified through sequencing but removed from the statistical analysis due to low representativeness. Lastly, the Proteobacteria phylum was reduced after the BG consumption, whose means of abundance were intermediate in treatments 0.07 and 0.28% and lower in treatment 0.14% BG. This result indicates a better protective role of BG at a dose of 0.14% since the main pathogenic bacteria, such as *Escherichia*, *Salmonella*, and *Campylobacter*, belong to the Proteobacteria phylum [65,66]. Furthermore, this phylum is correlated with inflammatory bowel disease in both humans and dogs [45,64].

Thus, considering the data and discussion regarding the fecal microbiota of the dogs in this study, it can be inferred that the inclusion of purified beta-1,3/1,6-glucan at all three doses resulted in a more favorable intestinal microbial environment.

Some studies that proposed to evaluate the influence of yeast cell wall supplementation reported different results regarding the digestibility of nutrients [35,36,37,60]. Although no negative result was detected, an increase in the crude protein ADC was observed after the consumption of diets 0.14 and 0.28% BG. This appears to be an unprecedented result for purified beta-1,3/1,6-glucans and contrasts with previous studies using the whole yeast cell wall [67,68]. However, it should be interpreted with caution since 0.14 and 0.28% BG treatments had a higher crude protein content.

A disparity in crude protein content was observed between the chemical compositions of the 0.14 (27.81%) and 0.28% (28.24%) BG diets, in contrast to the 0.0% (25.25%) and 0.07% (25.07%) BG diets. It is possible that during the extrusion process, protein source ingredients were included in greater concentration, a potential factor that resulted in an increase in the crude protein ADC of dogs.

It is known that prebiotics, by stimulating the proliferation of bacteria in the colon, can promote an increase in fecal nitrogen concentration and underestimate the crude protein ADC [69]. In a study by Pinna et al. [70], dogs that received a diet supplemented with fructooligosaccharides (FOS; 0.15%) and low protein content (24.1%) had higher crude protein ADCs when compared to animals fed a FOS-supplemented diet and high protein content (31.2%). In this case, it is speculated that the high concentration of undigested proteins, along with the prebiotic effect of FOS, favored a greater degree of proteolytic metabolism in the large intestine and, consequently, increased fecal nitrogen. Furthermore, it is known that certain genera of the families Erysipelotrichaceae, Clostridiaceae, and the *Peptococcus* genus are involved in intestinal proteolytic metabolism [48,71].

Bacteria involved in intestinal proteolytic metabolism are responsible for deaminating proteins and amino acids into ammonia, the main source of nitrogen in the microbiome [72]. In general, in situations where energy sources from carbohydrates are scarce, or there is an abundance in the passage of undigested proteins, such metabolism is stimulated, and consequently, there is an increase in the intestinal ammonia concentration [73]. Ammonia, together with other metabolites of protein fermentation called phenols, indoles, and BCFA, are considered putrefactive compounds and are responsible for the bad odor in feces [74]. The high concentration of intestinal ammonia is capable of causing damage to the mucosa and reducing the height of the intestinal villi, whose impact is the reduction in the absorptive capacity of the tissue [75] and, in addition, alters the cell-renewal cycle and may contribute to colorectal carcinogenesis [76].

The findings presented in this study indicate that the intensity of protein and amino acid metabolization into ammonia by the intestinal bacteria remained consisted, irrespective of the inclusion level of BG. This is a very common finding, which corroborates several other studies that evaluated the influence of yeast cell walls and prebiotic supplementation on intestinal ammonia concentration in healthy dogs [35,36,37,77,78].

Another important marker of intestinal health is the lactic acid concentration [79]. This metabolite results from carbohydrate fermentation by certain bacteria, such as *Lactobacillus* and *Bifidobacterium*, and exerts beneficial effects on the intestine by lowering the luminal pH and inhibiting the proliferation of harmful bacteria [72]. In this study, the absence of a positive effect regarding the intestinal lactate concentrations in the animals that consumed the experimental diets may be explained by an increase in bacterial species that use this metabolite in their own metabolism or by the low representativeness of *Lactobacillus* and *Bifidobacterium* [80]. The concentration of fecal lactic acid was determined in previous studies with other prebiotics, whose results corroborate the absence of difference for this variable [36,37,77,81].

No alterations were observed in the concentration of SCFA, BCFA, and total fatty acids after BG inclusion in the diets. This finding can be justified by two distinct hypotheses. The first is due to the rapid absorption of these compounds by the colonocytes, which reduces the fecal concentration, underestimates the detection and statistical comparison [77], and corroborates the results of other studies in the area [36,60,67,81]. SCFA are compounds considered beneficial, as they are important energy sources for the metabolism and proliferation of colonocytes, reduce the pH of the intestinal lumen, an unfavorable condition for pathogenic microorganisms, and have anti-inflammatory properties [77,82]. And the second hypothesis is related to the fact that purified beta-1,3/1,6-glucan has an immunomodulatory effect rather than a fermentative effect in the intestine. Thus, the modulation of fecal microbiota would be associated with the ability of these beta-glucans to interact with and activate macrophages, which can promote beneficial phagocytic activity within the intestinal microbial environment [83].

In summary, the digestive tract of animals cannot degrade beta-glucans. Thus, after its ingestion, one portion has a specific prebiotic effect on the intestinal microbiota, and another portion is captured in the small intestine by a cooperative process involving intestinal M cells and macrophages. The large beta-glucan particles are then internalized and fragmented into nanoparticles. During this process, they are transported to the bone marrow and the reticuloendothelial system, where the small fragments are released. These fragments are captured by other important players (cells) of the immune response. Finally, all cells involved in these processes are activated, and the biological response will be appropriately modulated [84].

Some studies correlated improvements in the immune status of dogs with purified beta-1,3/1,6-glucan supplementation [16,21,38,39]. This function derives from the ability of these compounds to stimulate both innate and adaptive immunity through the activation of neutrophils, B, T, and NK lymphocytes, and mainly macrophages [85,86]. In general, it is known that purified beta-1,3/1,6-glucan can stimulate or inhibit cytokine release by macrophages and modulate their phagocytic activity [87]. Through fermentation and the production of metabolites, the microbiota provides nutrients to the host and ensures the optimal functioning of tissues like gut-associated lymphoid tissue (GALT), considered the largest immune tissue of the organism [51].

The immunological variable results were similar between the treatments, which may reflect the healthy condition and absence of immunological challenges throughout the experiment. However, the relationship between CD4^+^ and CD8^+^ T cells was modulated through the ingestion of 0.14% BG. The CD4^+^:CD8^+^ index is an important marker of the efficiency of the immune system [88,89]. Elderly dogs showed immunological decline marked by a reduction in the relative population of CD4^+^ T lymphocytes and an increase in CD8^+^ T lymphocytes, which decreases the CD4^+^:CD8^+^ ratio [90]. Furthermore, Singh et al. [91] correlated low levels of CD4^+^:CD8^+^ with the occurrence of generalized demodicosis in dogs. This occurs because the population of T CD8^+^ only increases when there is an invasion of pathogens and should be reduced after controlling the infection [92]. When there is a constant increase, the body can be subjected to severe inflammatory reactions [93]. Therefore, the increase in this ratio between T CD4^+^ and T CD8^+^ lymphocytes suggests that 0.14% BG supplementation induces an increase in the adaptive response of healthy adult dogs.

These findings are important to guide further research aimed at deepening understanding of how beta-glucans interact with the gastrointestinal tract of dogs and other dietary components. A better understanding of such interactions can elucidate new dietary compositions that enhance the effects of beta-glucans. Therefore, future studies should use a greater number of animals kept supplemented for a longer period of time in order to obtain results focused on the responses brought by the currently recommended dose. Furthermore, more dose-response studies are also needed for other objectives and applications of BG.

## 5. Conclusions

Under the conditions under which this study was carried out and based on the results obtained, it can be concluded that the inclusion of purified beta-1,3/1,6-glucan does not reduce the digestibility of nutrients in the diet. Furthermore, the modulations observed in adaptive immunity (where 0.14% BG modulated the ratio of CD4+:CD8+ T lymphocytes), the beta diversity index, and the observation of beneficial bacterial phyla, families, and genera modulated after ingestion of 0.14% BG indicates that treatment at this dose (0.14% BG) elicited the most favorable response in dogs for the objectives of this proposed study.

## Figures and Tables

**Figure 1 microorganisms-12-00113-f001:**
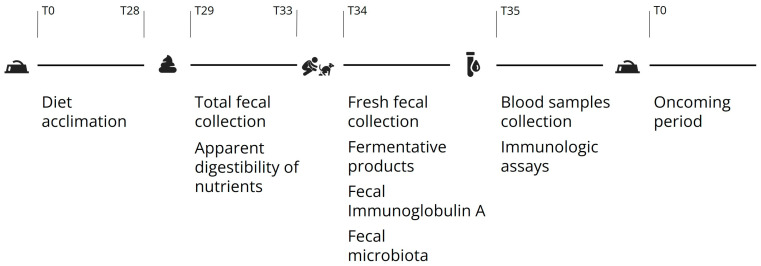
Procedures performed in each of the four experimental periods.

**Figure 2 microorganisms-12-00113-f002:**
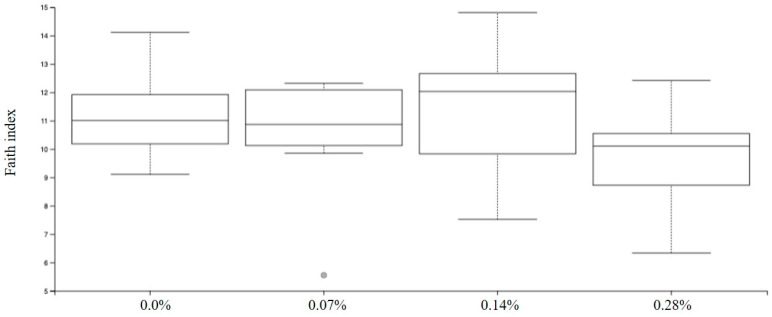
Faith’s phylogenetic diversity index observed in experimental groups.

**Figure 3 microorganisms-12-00113-f003:**
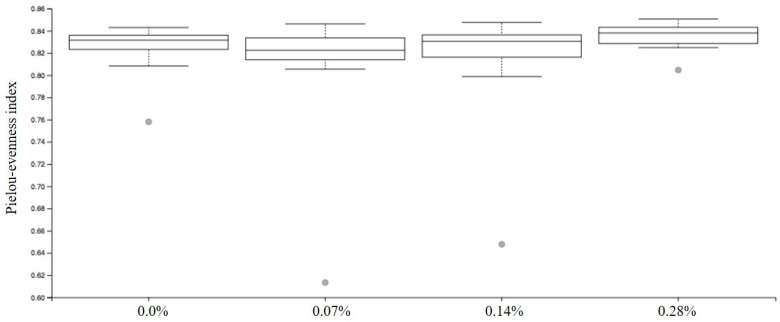
Pielou-evenness index observed in experimental groups.

**Figure 4 microorganisms-12-00113-f004:**
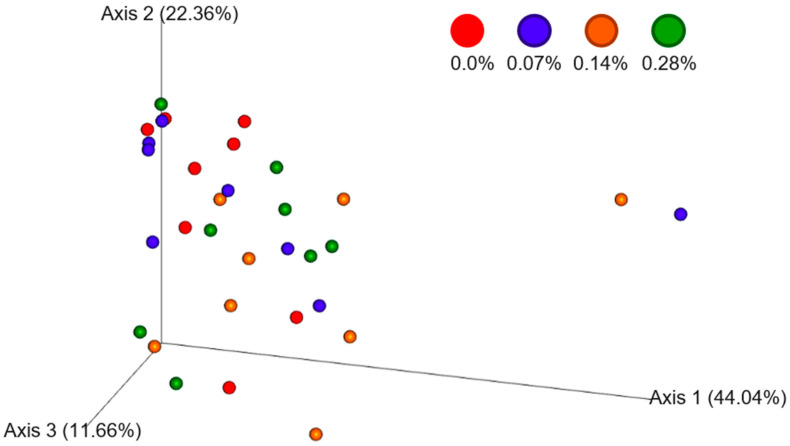
Three-dimensional principal coordinate analysis through relative data obtained through the sequencing of fecal samples from the experimental groups.

**Figure 5 microorganisms-12-00113-f005:**
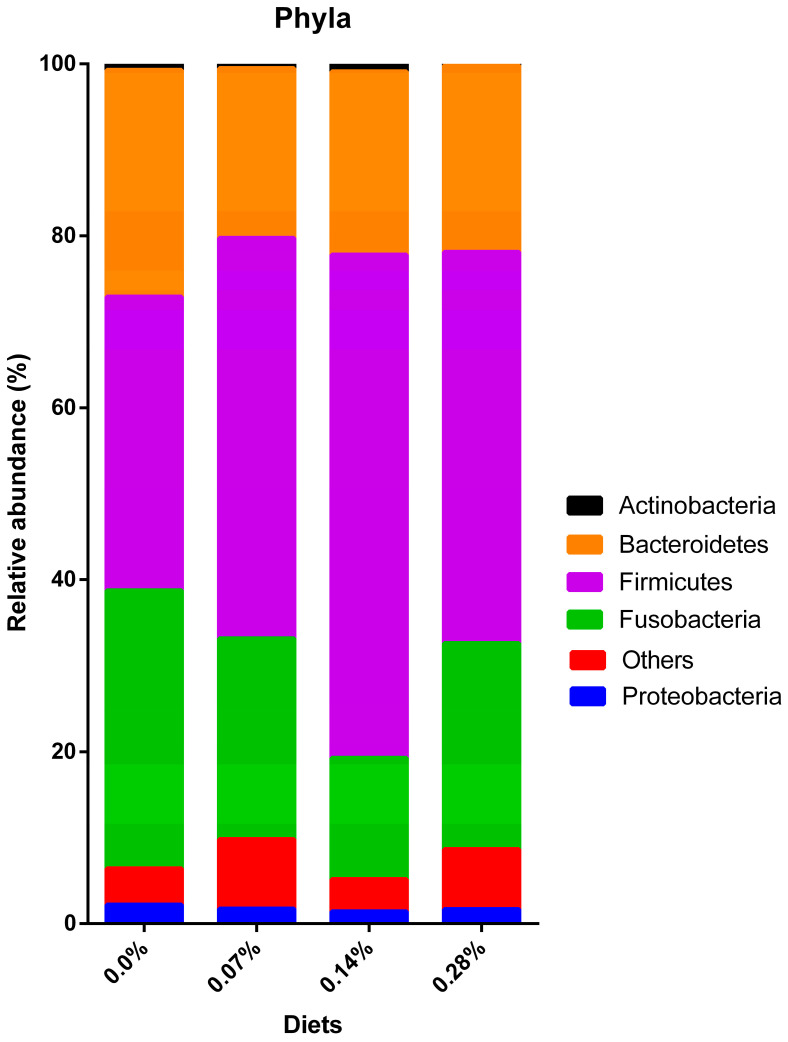
Distribution of bacterial phyla observed in experimental groups.

**Figure 6 microorganisms-12-00113-f006:**
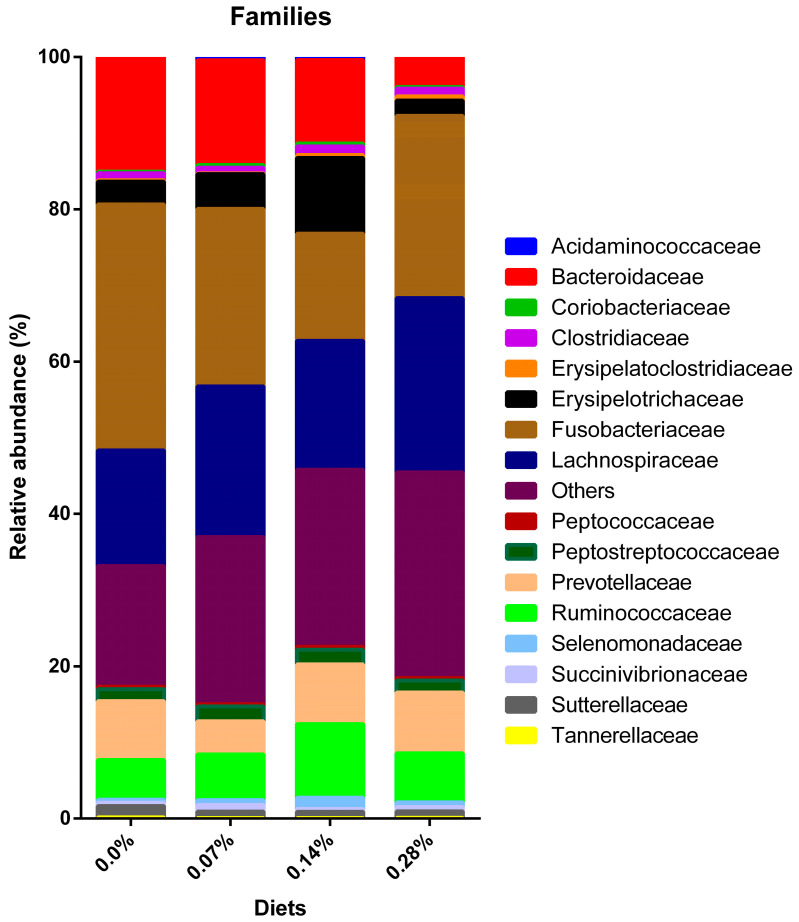
Distribution of bacterial families observed in experimental groups.

**Figure 7 microorganisms-12-00113-f007:**
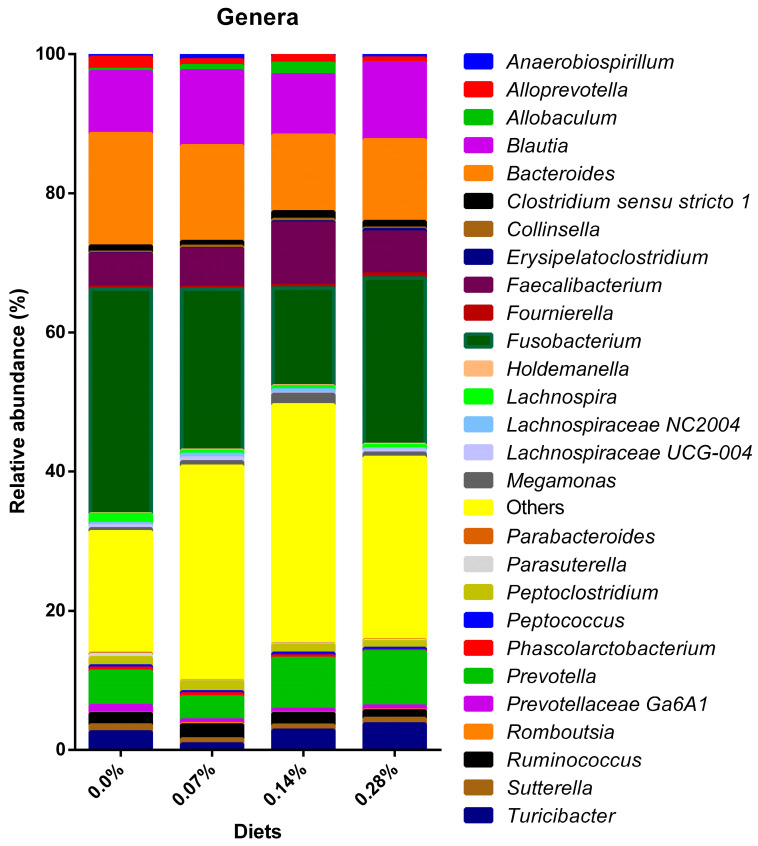
Distribution of bacterial genera observed in experimental groups.

**Figure 8 microorganisms-12-00113-f008:**
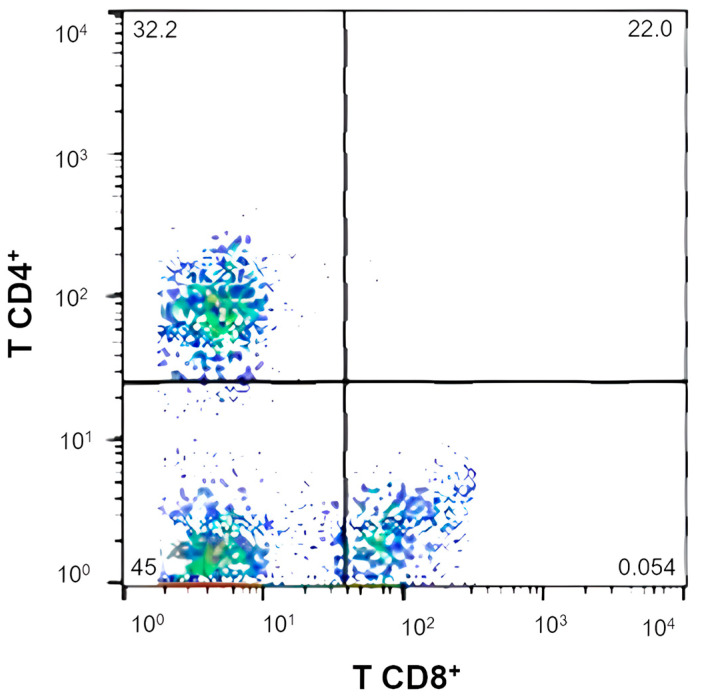
Lymphocyte T CD4^+^:CD8^+^ ratio of the experimental groups.

**Table 1 microorganisms-12-00113-t001:** Ingredient composition (%) of experimental foods.

Item	Diets			
	0.0%	0.07%	0.14%	0.28%
Ingredients (%)				
Corn grain	33.26	33.19	33.12	32.98
Common viscera meal	26.38	26.38	26.38	26.38
Broken rice	15.00	15.00	15.00	15.00
Corn gluten	7.99	7.99	7.99	7.99
Beet pulp	4.00	4.00	4.00	4.00
Fish oil	0.82	0.82	0.82	0.82
Potassium chloride	0.42	0.42	0.42	0.42
Mineral and vitamin premix ^1^	0.50	0.50	0.50	0.50
Common salt	0.30	0.30	0.30	0.30
Choline	0.17	0.17	0.17	0.17
Whole egg powder	0.15	0.15	0.15	0.15
Antifungal	0.10	0.10	0.10	0.10
Antioxidant	0.07	0.07	0.07	0.07
Methionine	0.03	0.03	0.03	0.03
Poultry viscera fat	6.81	6.81	6.81	6.81
Swine fat	4.00	4.00	4.00	4.00
Purified beta-1,3/1,6-glucan ^2^	0.00	0.07	0.14	0.28
Total	100	100	100	100

0.0% = dog-extruded dry food without beta-glucan inclusion; 0.07% = dog-extruded dry food with 0.07% beta-glucan inclusion; 0.14% = dog-extruded dry food with 0.14% beta-glucan inclusion; 0.28% = dog-extruded dry food with 0.28% beta-glucan inclusion; ^1^ Nutrient enrichment per kg: iron 131.57 mg, copper 16.56 mg, manganese 25.26 mg, zinc 135.83, iodine 1.4 mg, selenium 0.36 mg, vitamin A 18500 IU, vitamin E 134.56 mg, vitamin C 25.16 mg, vitamin D3 1295 IU, vitamin K, 2.66 mg, thiamine 12.26 mg, riboflavin 19.04 mg, pantothenic acid 32.3 mg, niacin 66.7 mg, pyridoxine 11.92 mg, folic acid 2.16 mg, biotin 0.44 mg, and cobalamin 0.81 mg; ^2^ Commercial product contained 60% purified beta 1,3/1,6-glucan in its composition (MacroGard^®^, Biorigin, Lençóis Paulista, Sao Paulo, Brazil).

**Table 2 microorganisms-12-00113-t002:** Chemical composition of experimental foods.

Item	Diets			
	0.0%	0.07%	0.14%	0.28%
Dry matter (%)	93.11	94.31	94.00	93.13
	Chemical composition in DM (%)			
Organic matter	92.13	92.07	92.04	91.93
Crude protein	25.25	25.07	27.81	28.24
Fat	17.69	17.82	17.71	17.42
Ash	7.87	7.93	7.96	8.07
Crude fiber	10.17	9.92	10.15	8.28
Nitrogen-free extract ^1^	39.02	39.26	36.37	37.99
Calcium	2.09	2.07	2.13	2.06
Phosphorus	1.19	1.17	1.18	1.17
Metabolizable energy (kcal/g) ^2^	4.10	4.15	4.14	4.09

0.0% = dog-extruded dry food without beta-glucan inclusion; 0.07% = dog-extruded dry food with 0.07% beta-glucan inclusion; 0.14% = dog-extruded dry food with 0.14% beta-glucan inclusion; 0.28% = dog-extruded dry food with 0.28% beta-glucan inclusion. DM = dry matter. ^1^ Nitrogen-free extract was calculated by the difference of the known macronutrient content. ^2^ Metabolizable energy was estimated according to NRC [32].

**Table 3 microorganisms-12-00113-t003:** Means and standard errors of the relative abundances of the phyla observed in the experimental groups.

Phyla (%)	Diets				*p*
	0.0%	0.07%	0.14%	0.28%	
Actinobacteria	0.72 ± 0.49 ^a^	0.51 ± 0.35 ^b^	0.90 ± 0.61 ^a^	0.15 ± 0.10 ^c^	<0.0001
Bacteroidetes	26.38 ± 6.99 ^a^	19.75 ± 5.70 ^c^	21.30 ± 6.03 ^b^	21.73 ± 6.12 ^b^	<0.0001
Firmicutes	34.18 ± 9.16 ^c^	46.59 ± 10.13 ^b^	58.54 ± 9.88 ^a^	45.54 ± 10.10 ^b^	<0.0001
Fusobacteria	32.32 ± 5.76 ^a^	23.33 ± 4.72 ^b^	14.09 ± 3.19 ^c^	23.92 ± 4.80 ^b^	<0.0001
Proteobacteria	2.21 ± 1.08 ^a^	1.72 ± 0.84 ^b^	1.42 ± 0.69 ^c^	1.67 ± 0.82 ^b^	<0.0001

0.0% = dog-extruded dry food without beta-glucan inclusion; 0.07% = dog-extruded dry food with 0.07% beta-glucan inclusion; 0.14% = dog-extruded dry food with 0.14% beta-glucan inclusion; 0.28% = dog-extruded dry food with 0.28% beta-glucan inclusion. ^a–c^ Averages in the same line followed by different letters differed by 1% in the Tukey–Kramer test adjusted by PROC MIXED.

**Table 4 microorganisms-12-00113-t004:** Means and standard errors of relative abundances of families observed in experimental groups.

Families (%)	Diets				*p*
	0.0%	0.07%	0.14%	0.28%	
Acidaminococcaceae	0.46 ± 0.09 ^a^	0.42 ± 0.09 ^a^	0.38 ± 0.08 ^a^	0.16 ± 0.04 ^b^	<0.0001
Bacteroidaceae	16.17 ± 3.53 ^a^	13.77 ± 3.09 ^b^	10.99 ± 2.55 ^c^	11.76 ± 2.70 ^c^	<0.0001
Clostridiaceae	0.94 ± 0.54 ^ab^	0.74 ± 0.43 ^b^	1.14 ± 0.65 ^a^	1.02 ± 0.59 ^ab^	0.0011
Coriobacteriaceae	0.20 ± 0.10 ^b^	0.35 ± 0.17 ^a^	0.37 ± 0.18 ^a^	0.22 ± 0.11 ^b^	0.0111
Erysipelatoclostridiaceae	0.17 ± 0.07 ^b^	0.10 ± 0.04 ^b^	0.37 ± 0.16 ^a^	0.54 ± 0.23 ^a^	<0.0001
Erysipelotrichaceae	2.99 ± 1.72 ^c^	4.56 ± 2.58 ^b^	9.94 ± 5.29 ^a^	2.02 ± 1.17 ^d^	<0.0001
Fusobacteriaceae	32.32 ± 5.49 ^a^	23.33 ± 4.49 ^b^	14.09 ± 3.04 ^c^	23.92 ± 4.57 ^b^	<0.0001
Lachnospiraceae	15.18 ± 4.62 ^d^	19.77 ± 5.69 ^b^	16.91 ± 5.04 ^c^	22.89 ± 6.33 ^a^	<0.0001
Peptococcaceae	0.31 ± 0.09	0.30 ± 0.09	0.37 ± 0.11	0.32 ± 0.09	0.3061
Peptostreptococcaceae	1.57 ± 0.35 ^c^	1.95 ± 0.43 ^a^	1.89 ± 0.42 ^ab^	1.58 ± 0.35 ^bc^	0.0003
Prevotellaceae	7.74 ± 3.04 ^a^	4.37 ± 1.78 ^b^	7.85 ± 3.08 ^a^	7.99 ± 3.13 ^a^	<0.0001
Ruminococcaceae	5.20 ± 1.11 ^c^	5.98 ± 1.27 ^b^	9.66 ± 1.97 ^a^	6.44 ± 1.36 ^b^	<0.0001
Selenomonadaceae	0.41 ± 0.20 ^c^	0.64 ± 0.32 ^b^	1.49 ± 0.73 ^a^	0.60 ± 0.30 ^b^	<0.0001
Succinivibrionaceae	0.45 ± 0.14 ^b^	0.86 ± 0.26 ^a^	0.39 ± 0.12 ^b^	0.59 ± 0.18 ^ab^	<0.0001
Sutterellaceae	1.47 ± 0.96 ^a^	0.86 ± 0.57 ^b^	0.82 ± 0.53 ^b^	0.86 ± 0.57 ^b^	<0.0001
Tannerellaceae	0.16 ± 0.03 ^a^	0.07 ± 0.00 ^b^	0.06 ± 0.00 ^c^	0.09 ± 0.00 ^a^	0.0438

0.0% = dog-extruded dry food without beta-glucan inclusion; 0.07% = dog-extruded dry food with 0.07% beta-glucan inclusion; 0.14% = dog-extruded dry food with 0.14% beta-glucan inclusion; 0.28% = dog-extruded dry food with 0.28% beta-glucan inclusion. ^a–d^ Averages in the same line followed by different letters differed by 5% in the Tukey–Kramer test adjusted by PROC MIXED.

**Table 5 microorganisms-12-00113-t005:** Means and standard errors of the relative abundances of genera observed in experimental groups.

Genera (%)	Diets				*p*
	0.0%	0.07%	0.14%	0.28%	
*Allobaculum*	0.32 ± 1.76 ^c^	0.73 ± 4.03 ^b^	1.65 ± 9.03 ^a^	0.01 ± 0.03 ^d^	<0.0001
*Alloprevotella*	1.78 ± 0.48 ^a^	0.84 ± 0.23 ^c^	1.09 ± 0.30 ^b^	0.69 ± 0.19 ^c^	<0.0001
*Anaerobiospirillum*	0.44 ± 0.14 ^b^	0.83 ± 0.26 ^a^	0.39 ± 0.12 ^b^	0.58 ± 0.18 ^ab^	0.0002
*Bacteroides*	16.17 ± 3.53 ^a^	13.77 ± 3.09 ^b^	10.99 ± 2.55 ^c^	11.76 ± 2.70 ^c^	<0.0001
*Blautia*	8.91 ± 2.49 ^b^	10.78 ± 2.96 ^a^	8.71 ± 2.44 ^b^	11.03 ± 3.02 ^a^	<0.0001
*Clostridium sensu stricto 1*	0.88 ± 0.50 ^ab^	0.71 ± 0.40 ^b^	1.09 ± 0.61 ^a^	0.96 ± 0.54 ^ab^	0.0012
*Collinsella*	0.20 ± 0.10 ^b^	0.35 ± 0.17 ^a^	0.37 ± 0.18 ^a^	0.22 ± 0.11 ^b^	0.0111
*Erysipelatoclostridium*	0.06 ± 0.02 ^b^	0.06 ± 0.03 ^b^	0.15 ± 0.06 ^ab^	0.36 ± 0.11 ^a^	0.0008
*Faecalibacterium*	4.76 ± 1.06 ^c^	5.48 ± 1.21 ^b^	9.01 ± 1.91 ^a^	6.03 ± 1.32 ^b^	<0.0001
*Fournierella*	0.26 ± 0.06 ^ab^	0.24 ± 0.06 ^b^	0.33 ± 0.08 ^ab^	0.55 ± 0.14 ^a^	0.0075
*Fusobacterium*	32.33 ± 5.46 ^a^	23.18 ± 4.45 ^b^	14.09 ± 3.03 ^c^	23.94 ± 4.55 ^b^	<0.0001
*Holdemanella*	0.08 ± 0.03	0.17 ± 0.06	0.14 ± 0.04	0.17 ± 0.06	0.2969
*Lachnospira*	1.30 ± 0.56 ^a^	0.41 ± 0.18 ^b^	0.42 ± 0.18 ^b^	0.44 ± 0.19 ^b^	0.0002
*Lachnospiraceae NC2004*	0.32 ± 0.11 ^b^	0.52 ± 0.19 ^a^	0.51 ± 0.18 ^a^	0.22 ± 0.08 ^b^	<0.0001
*Lachnospiraceae UCG-004*	0.44 ± 0.09 ^a^	0.55 ± 0.12 ^a^	0.16 ± 0.04 ^b^	0.43 ± 0.09 ^a^	<0.0001
*Megamonas*	0.41 ± 0.20 ^c^	0.64 ± 0.32 ^b^	1.48 ± 0.72 ^a^	0.60 ± 0.30 ^b^	<0.0001
*Parabacteroides*	0.16 ± 0.00 ^a^	0.07 ± 0.00 ^bc^	0.07 ± 0.00 ^c^	0.09 ± 0.00 ^b^	0.0191
*Parasuterella*	0.49 ± 0.14 ^a^	0.12 ± 0.04 ^b^	0.15 ± 0.05 ^b^	0.12 ± 0.04 ^b^	0.0002
*Peptoclostridium*	1.15 ± 0.30 ^ab^	1.41 ± 0.37 ^a^	1.13 ± 0.30 ^b^	0.98 ± 0.26 ^b^	<0.0001
*Peptococcus*	0.31 ± 0.09	0.30 ± 0.09	0.37 ± 0.11	0.32 ± 0.09	0.3061
*Phascolarctobacterium*	0.46 ± 0.09 ^a^	0.42 ± 0.09 ^a^	0.38 ± 0.08 ^a^	0.16 ± 0.04 ^b^	<0.0001
*Prevotella*	4.94 ± 2.12 ^b^	3.34 ± 1.46 ^c^	7.32 ± 3.06 ^a^	7.81 ± 3.25 ^a^	<0.0001
*Prevotellaceae Ga6A1*	1.11 ± 0.37 ^a^	0.52 ± 0.18 ^b^	0.47 ± 0.16 ^b^	0.58 ± 0.20 ^b^	<0.0001
*Romboutsia*	0.06 ± 0.03	0.18 ± 0.10	0.12 ± 0.05	0.10 ± 0.04	0.2994
*Sutterella*	1.04 ± 0.70 ^a^	0.75 ± 0.51 ^b^	0.73 ± 0.50 ^b^	0.77 ± 0.52 ^b^	0.0002
*Turicibacter*	2.56 ± 0.83 ^b^	0.83 ± 0.28 ^c^	2.82 ± 0.91 ^b^	3.74 ± 1.20 ^a^	<0.0001
*Ruminococcus*	1.58 ± 0.46 ^b^	2.00 ± 0.58 ^a^	1.67 ± 0.49 ^ab^	1.10 ± 0.32 ^c^	<0.0001
*Uncultured*	1.25 ± 0.46 ^c^	1.45 ± 0.53 ^c^	1.97 ± 0.71 ^b^	2.44 ± 0.87 ^a^	<0.0001

0.0% = dog-extruded dry food without beta-glucan inclusion; 0.07% = dog-extruded dry food with 0.07% beta-glucan inclusion; 0.14% = dog-extruded dry food with 0.14% beta-glucan inclusion; 0.28% = dog-extruded dry food with 0.28% beta-glucan inclusion. ^a–d^ Averages in the same line followed by different letters differed by 5% in the Tukey–Kramer test adjusted by PROC MIXED.

**Table 6 microorganisms-12-00113-t006:** Apparent digestibility coefficients of nutrients (%) of experimental diets.

Item (%)	Diets				SEM	*p*
	0.0%	0.07%	0.14%	0.28%		
Dry matter	82.25	82.75	82.72	83.29	0.44	0.3162
Organic matter	87.83	88.32	88.34	88.71	0.31	0.1546
Crude protein	85.85 ^b^	85.99 ^b^	87.50 ^a^	88.19 ^a^	0.33	<0.0001
Ethereal extract	96.58	96.20	96.26	96.52	0.26	0.3703
Ash	69.47	81.55	77.89	79.09	5.41	0.4065
Crude fiber	17.01	18.11	17.75	21.52	2.15	0.3613
Nitrogen-free extract	87.17	87.94	88.04	87.61	0.53	0.5585

0.0% = dog-extruded dry food without beta-glucan inclusion; 0.07% = dog-extruded dry food with 0.07% beta-glucan inclusion; 0.14% = dog-extruded dry food with 0.14% beta-glucan inclusion; 0.28% = dog-extruded dry food with 0.28% beta-glucan inclusion. SEM = standard error of the mean. ^a–b^ Averages in the same line followed by different letters differed by 5% in the Tukey–Kramer test adjusted by PROC MIXED.

**Table 7 microorganisms-12-00113-t007:** Fermentative products, fatty acids, and fecal pH.

Item	Diets				SEM	*p*
	0.0%	0.07%	0.14%	0.28%		
	Fermentative products					
Ammoniacal nitrogen (mmol/kg of DM)	6.97	6.92	7.28	7.13	0.62	0.9576
Lactic acid (mmol/kg of DM)	4.80	5.41	5.65	5.21	0.72	0.5905
Total fatty acids (mmol/kg of DM)	54.66	54.18	58.26	56.15	2.40	0.3792
Fecal pH	6.68	6.66	6.53	6.59	0.12	0.7406
	Short-chain fatty acids (mmol/kg of DM)					
Acetic acid	33.13	31.75	32.98	32.91	1.77	0.8785
Propionic acid	13.77	14.34	15.64	15.13	1.00	0.1219
Butyric acid	6.26	6.58	7.72	6.57	1.00	0.6592
	Branched-chain fatty acids (mmol/kg of DM)					
Valeric acid	0.10	0.29	0.18	0.11	0.09	0.4069
Isovaleric acid	0.76	0.69	1.00	0.81	0.10	0.0664
Isobutyric acid	0.66	0.61	0.76	0.65	0.06	0.1466
Total BCFA	1.49	1.52	1.92	1.54	0.18	0.1534

0.0% = dog-extruded dry food without beta-glucan inclusion; 0.07% = dog-extruded dry food with 0.07% beta-glucan inclusion; 0.14% = dog-extruded dry food with 0.14% beta-glucan inclusion; 0.28% = dog-extruded dry food with 0.28% beta-glucan inclusion. DM = dry matter. SEM = standard error of the mean. BCFA = branched-chain fatty acids.

**Table 8 microorganisms-12-00113-t008:** Results of lymphocyte immunophenotyping tests (percentage of T CD4^+^ and T CD8^+^ cells and T CD4^+^:CD8^+^ ratio), phagocytosis, oxidative burst, and fecal IgA.

Item	Diets				SEM	*p*
	0.0%	0.07%	0.14%	0.28%		
Phagocytosis (%)	68.63	75.25	76.79	78.04	4.15	0.0692
Phagocytosis intensity	32.85	36.56	30.33	37.80	7.47	0.2546
Oxidative burst PMA	894.00	809.37	785.62	1029.12	160.39	0.4423
T CD4^+^ lymphocytes (%)	37.9	42.49	44.53	42.60	2.45	0.2210
T CD8^+^ lymphocytes (%)	29.81	27.33	23.35	26.23	2.48	0.1237
T CD4^+^:CD8^+^ ratio	1.33 ^b^	1.71 ^ab^	1.99 ^a^	1.74 ^ab^	0.19	0.0368
Fecal IgA fecal (pg/mL)	225.091	271.41	242.35	229.73	53.81	0.9254

0.0% = dog-extruded dry food without beta-glucan inclusion; 0.07% = dog-extruded dry food with 0.07% beta-glucan inclusion; 0.14% = dog-extruded dry food with 0.14% beta-glucan inclusion; 0.28% = dog-extruded dry food with 0.28% beta-glucan inclusion. SEM = standard error of the mean; PMA = phorbol myristate acetate. ^a–b^ Averages in the same line followed by different letters differed by 5% in the Tukey–Kramer test adjusted by PROC MIXED.

## Data Availability

The original contributions presented in the study are included in the article, further inquiries can be directed to the corresponding author.

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
