# Peer review of "Effects of Increasing Levels of Purified Beta-1,3/1,6-Glucans on the Fecal Microbiome, Digestibility, and Immunity Variables of Healthy Adult Dogs"

_microorganisms, 2024, doi:10.3390/microorganisms12010113_

Round 1
Reviewer 1 Report
Comments and Suggestions for Authors
The article titled "Effects of increasing levels of purified beta-13/16-glucans on the fecal microbiome digestibility and immunity variables of healthy adult dogs" by Pedro Marchi et al. explores the impact of various concentrations of beta-13/16-glucans (BG) in dog diets on various health parameters. The study was conducted in accordance with ethical standards and approved by an ethics committee​.
Here is a rational criticism of the article:
-
1) Research Design and Sample Size: The study used a balanced 4x4 Latin square design with eight dogs (four border collies and four English cocker spaniels). While this design helps control variability and allows for multiple treatments, the small sample size could limit the generalizability of the findings. A larger sample size with a more diverse range of breeds could provide more comprehensive insights into the effects of BG across different dog populations.
* While this might be a financial burden, it is recommended to include "Justification for Animal Data" (optional - Power Analysis grahic with explanation) -
2) Dietary Composition and Control: The diets prepared contained varying levels of BG (0.0%, 0.07%, 0.14%, and 0.28%). Although the study controlled for energy intake and other dietary components, the intricate interactions between BG and other dietary elements were not fully explored.
* Please include comment why those specific concentrations were used.
* Please discuss how future research could benefit from examining these interactions to better understand the holistic impact of BG-enriched diets on dog health​. -
3) Measurable Outcomes and Statistical Analysis: The study reported no significant differences in most nutrients' apparent digestibility coefficients (ADCs), except for crude protein, which showed higher digestibility at 0.14% and 0.28% BG levels. While these findings are valuable, the study could have been strengthened by a more detailed statistical analysis, particularly regarding the potential effects of BG on other health-related variables, such as immunity and gut microbiota composition​.
* As an alternative - limitations could be discussed in "Discussion" section. -
4) Long-Term Effects and Practical Implications: The study provides initial insights into the potential benefits of BG in dog diets. However, it does not address long-term effects or the practical implications of incorporating BG into commercial dog food.
* Please discuss how future research could focus on long-term health impacts and the feasibility of mass-producing BG-enriched dog food. -
5) Broader Contextualization: The study is somewhat limited in its discussion of how these findings fit into the broader context of canine nutrition and gut health. Expanding this discussion could help readers understand the practical implications of the research and how it contributes to the existing body of knowledge in canine nutrition.
Overall, the study provides valuable information about the effects of BG on dogs' gut health and nutrition. However, its findings could be bolstered by a larger and more diverse sample size, deeper exploration of dietary interactions, long-term effect studies, and a broader contextual discussion within the field of canine health and nutrition.
Best wishes to the authors in their future endeavours.
Author Response
Dear Editor,
The authors wish to thank you for the attention given to our research. We have considered all the suggestions and comments by the reviewers, which are addressed below. Thank you once again for considering our work for publishing. I hope you are also keeping safe and well.
Kind regards,
Prof. Dr. Thiago Henrique Annibale Vendramini
Reviewer 1
The article titled "Effects of increasing levels of purified beta-13/16-glucans on the fecal microbiome digestibility and immunity variables of healthy adult dogs" by Pedro Marchi et al. explores the impact of various concentrations of beta-13/16-glucans (BG) in dog diets on various health parameters. The study was conducted in accordance with ethical standards and approved by an ethics committee​.
Here is a rational criticism of the article:
1) Research Design and Sample Size: The study used a balanced 4x4 Latin square design with eight dogs (four border collies and four English cocker spaniels). While this design helps control variability and allows for multiple treatments, the small sample size could limit the generalizability of the findings. A larger sample size with a more diverse range of breeds could provide more comprehensive insights into the effects of BG across different dog populations.
* While this might be a financial burden, it is recommended to include "Justification for Animal Data" (optional - Power Analysis grahic with explanation)
Author: Thank you for your critical point and consideration. Latin squares are widely employed in animal nutrition essays, both in ruminant and monogastric nutrition. This design allows for the addition of a reasonable number of degrees of freedom to the residual (in this case, 18 degrees of freedom applied to the residual), while simultaneously controlling for variations in period, animal, and square. The virtue of this design lies in the fact that the experimental units consist of the combination of an animal in each period and treatment, totaling 32 experimental units in our study (8 for treatment).
2) Dietary Composition and Control: The diets prepared contained varying levels of BG (0.0%, 0.07%, 0.14%, and 0.28%). Although the study controlled for energy intake and other dietary components, the intricate interactions between BG and other dietary elements were not fully explored.
*Please include comment why those specific concentrations were used.
*Please discuss how future research could benefit from examining these interactions to better understand the holistic impact of BG-enriched diets on dog health​.
Author: We have added a sentence to the "2.2 Diets and experimental design" section, which rationalizes the use of these doses and references previous studies.
We have added a discussion on the second topic at the end of the discussion section, in a paragraph addressing the main limitations raised by the reviewer. Thank you.
3) Measurable Outcomes and Statistical Analysis: The study reported no significant differences in most nutrients' apparent digestibility coefficients (ADCs), except for crude protein, which showed higher digestibility at 0.14% and 0.28% BG levels. While these findings are valuable, the study could have been strengthened by a more detailed statistical analysis, particularly regarding the potential effects of BG on other health-related variables, such as immunity and gut microbiota composition​.
* As an alternative - limitations could be discussed in "Discussion" section.
Author: We have added a discussion on this topic at the end of the discussion section, in a paragraph addressing the main limitations raised by the reviewer. Thank you.
4) Long-Term Effects and Practical Implications: The study provides initial insights into the potential benefits of BG in dog diets. However, it does not address long-term effects or the practical implications of incorporating BG into commercial dog food.
* Please discuss how future research could focus on long-term health impacts and the feasibility of mass-producing BG-enriched dog food.
Author: We have added a discussion on this topic at the end of the discussion section, in a paragraph addressing the main limitations raised by the reviewer. Thank you.
5) Broader Contextualization: The study is somewhat limited in its discussion of how these findings fit into the broader context of canine nutrition and gut health. Expanding this discussion could help readers understand the practical implications of the research and how it contributes to the existing body of knowledge in canine nutrition.
Overall, the study provides valuable information about the effects of BG on dogs' gut health and nutrition. However, its findings could be bolstered by a larger and more diverse sample size, deeper exploration of dietary interactions, long-term effect studies, and a broader contextual discussion within the field of canine health and nutrition.
Best wishes to the authors in their future endeavours.
Author: We would like to express our gratitude for all the considerations, comments, and suggestions. We hope that our changes meet your expectations.

Reviewer 2 Report
Comments and Suggestions for Authors
Manuscript is constructed at a very good level. Introduction is well written but it misses some bacterial species of microbiota characteristics for dogs and change of microbiota after Beta glucans application.
The material and method is missing a deeper description of beta glucans used in study.
In table 1, 2 and other is percentuali described beta glucans? If yes it is necessary to be mentioned in the table. Or in table designation that belongs to the explanatory notes below the table.
Families of microorganisms are without italics.
The meaning of prebiotic is first mentioned in the results. Why is not mentioned in the Introduction.
The discussion is very well described but the conclusion needs more information about results in the study.
In the abstract is missing some information about most identified families of bacteria.
Comments on the Quality of English LanguageEnglish language is in good level.
Author Response
Dear Editor,
The authors wish to thank you for the attention given to our research. We have considered all the suggestions and comments by the reviewers, which are addressed below. Thank you once again for considering our work for publishing. I hope you are also keeping safe and well.
Kind regards,
Prof. Dr. Thiago Henrique Annibale Vendramini
Reviewer 2
Manuscript is constructed at a very good level. Introduction is well written but it misses some bacterial species of microbiota characteristics for dogs and change of microbiota after beta-glucans application.
Author: A sentence highlighting the main microorganisms present in the intestinal microbiome of dogs has been added. However, there are no studies that have assessed changes in the fecal microbiota of dogs after the consumption of purified beta-1,3/1,6-glucans, only yeast coproducts (which contain other components such as mannanoligosaccharides). Thank you.
The material and method is missing a deeper description of beta glucans used in study.
Author: The information was included in the requested location. Thank you.
In table 1, 2 and other is percentuali described beta glucans? If yes, it is necessary to be mentioned in the table. Or in table designation that belongs to the explanatory notes below the table.
Author: All footnotes contain explanatory notes regarding the percentage of beta-glucans. Thank you.
Families of microorganisms are without italics.
Author: All families of microorganisms are italicized. Thank you.
The meaning of prebiotic is first mentioned in the results. Why is not mentioned in the Introduction.
Author: Thank you for the suggestion and we've included the meaning for introduction.
The discussion is very well described but the conclusion needs more information about results in the study.
Author: We have included more information about the results of the study in the conclusion. Thank you.
In the abstract is missing some information about most identified families of bacteria.
Author: We have included additional details regarding the observed modulations at the phylum, genus, and family abundance. Thank you.

Reviewer 3 Report
Comments and Suggestions for Authors
The manuscript (microorganisms-2734186) focuses on the evaluation of the effects of different levels of purified beta-1,3/1,6-glucans in diet on digestibility, immunity, and fecal microbiota of adult dogs. Overall, the manuscript provides valuable insights into the effects of purified beta-1,3/1,6-glucans levels on dogs' health. However, there are several points that need to be addressed as follows:
1) In the abstract, include more information about the obtained results to better represent the study's findings.
2) Line 55; Can you provide more details on the production of purified beta-1,3/1,6-glucan (BG) from Saccharomyces cerevisiae?
3) Lines 57-59; Could you provide additional information on the mechanisms of action of BG?
4) The hypothesis of the study should be clarified at the end of the introduction section.
5) Lines 76-78; Could you provide some information regarding the overall weight of dogs?
6) Lines 91-92; The authors had to justify on what basis they selected the level of BG.
7) How were the BG levels added to the diet?
8) The source of BG (product, concentration, company, city, country) should be clearly stated in Section 2.2.
9) Section 2.3; The instruments and chemicals used must contain all full information such as model, company, city, and country.
10) Line 154; What method is employed to gather the entirety of daily fecal matter?
11) Line 297; Please include "NRC (2006)" in the list of references.
12) How was 0.14% BG computed as the optimum level?
Comments on the Quality of English Language
--
Author Response
Dear Editor,
The authors wish to thank you for the attention given to our research. We have considered all the suggestions and comments by the reviewers, which are addressed below. Thank you once again for considering our work for publishing. I hope you are also keeping safe and well.
Kind regards,
Prof. Dr. Thiago Henrique Annibale Vendramini
Reviewer 3
The manuscript (microorganisms-2734186) focuses on the evaluation of the effects of different levels of purified beta-1,3/1,6-glucans in diet on digestibility, immunity, and fecal microbiota of adult dogs. Overall, the manuscript provides valuable insights into the effects of purified beta-1,3/1,6-glucans levels on dogs' health. However, there are several points that need to be addressed as follows:
1) In the abstract, include more information about the obtained results to better represent the study's findings.
Author: Thank you for your request. Unfortunately, the journal's guidelines do not allow the abstract to exceed 200 words. Our abstract was written to align as closely as possible with the journal's standards, ensuring that the required information is distributed throughout the manuscript. However, we have included additional details regarding the observed modulations at the phylum, genus, and family abundance and one consideration about the T CD4+:CD8+ ratio.
2) Line 55; Can you provide more details on the production of purified beta-1,3/1,6-glucan (BG) from Saccharomyces cerevisiae?
Author: The information was included in the “2.2. Diets and experimental design” section. Thank you.
3) Lines 57-59; Could you provide additional information on the mechanisms of action of BG?
Author: The digestive tract of animals cannot degrade beta-glucans. Thus, after its ingestion, one portion has specific prebiotic effect for the intestinal microbiota and another portion is captured in the small intestine by a cooperative process involving intestinal M cells and macrophages. The large beta-glucan particles are then internalized and fragmented into nanoparticles. During this process, they are transported to the bone marrow and to reticuloendothelial system, where the small fragments are released. These fragments are captured by other important players (cells) of the immune response. Finally, all cells involved in these processes are activated and the biological response will be appropriately modulated [1].
- Chan, G. C.; Chan, W. K.; Sze, D. M. The effects of beta-glucan on human immune and cancer cells. Hematol. Oncol. 2009, 2(25), 1-11. doi:10.1186/1756-8722-2-25.
This information was included in the discussion. Thank you.
4) The hypothesis of the study should be clarified at the end of the introduction section.
Author: The study hypothesis was clarified and included at the end of the introduction. Thank you.
5) Lines 76-78; Could you provide some information regarding the overall weight of dogs?
Author: The information was included in the requested location. Thank you.
6) Lines 91-92; The authors had to justify on what basis they selected the level of BG.
Author: The levels used were based on some previous studies (not dose-response) that proposed evaluating the influence of β-glucan supplementation, these studies showed evident effects, and thus an intermediate dose was established and then defined the other doses with equidistant points. This information was included in the manuscript.
- de Oliveira, C.A.F.; Vetvicka, V.; Zanuzzo, F.S. β-Glucan successfully stimulated the immune system in different jawed vertebrate species. Comp Immunol Microbiol Infect Dis. 2019, 62, 1–6. doi: 10.1016/j.cimid.2018.11.006
- Vetvicka, V.; de Oliveira, C.A.F. β(1-3)(1-6)-D-glucans Modulate Immune Status and Blood Glucose Levels in Dogs. Br J Pharm Med Res. 2014, 4, 981–991. doi: 10.9734/BJPR/2014/7862
- Rychlik, A.; Nieradka, R.; Kander, M.; Nowicki, M.; Wdowiak, M.; Sawerska, A.K. The effectiveness of natural and synthetic immunomodulators in the treatment of inflammatory bowel disease in dogs. Acta Vet Hung. 2013, 61, 297–308. doi: 10.1556/AVet.2013.015
7) How were the BG levels added to the diet?
Author: The BG were added to dog food prior to extrusion process. This information was included in the text. Thank you.
8) The source of BG (product, concentration, company, city, country) should be clearly stated in Section 2.2.
Author: The requested information has been included, thank you.
9) Section 2.3; The instruments and chemicals used must contain all full information such as model, company, city, and country.
Author: The information was included as requested, thank you.
10) Line 154; What method is employed to gather the entirety of daily fecal matter?
Author: Dietary apparent digestibility coefficients (ADC) of nutrients were determined by the total fecal collection method according to AAFCO (2019). This information has been included.
- Association of American Feed Control Officials; AAFCO:Washington, DC, USA, 2019.
11) Line 297; Please include "NRC (2006)" in the list of references.
Author: Sorry, the reference was included, thank you.
12) How was 0.14% BG computed as the optimum level?
Author: Precisely the 0.14% dose was best evaluated through the results obtained in this study, where the modulations observed in adaptive immunity (where 0.14% of BG modulated the proportion of CD4+:CD8+ T lymphocytes), beta diversity index and an observation of beneficial bacterial phyla, families and genera modulated after ingestion of 0.14% BG indicate that treatment at this dose (0.14% BG) elicited the most favorable response in dogs. Therefore, we understand that it was the best dose evaluated, but not necessarily the ideal dose. In fact, the ideal doses must be defined according to the purpose of application of the BG (for example prebiotic function, satiety immune function and so on).
Round 2
Reviewer 2 Report
Comments and Suggestions for Authors
Line 28 Family is again with italics, author need think if describing new part of manuscript is necessary allocate previous comments into new parts.
Author Response
Sorry, we made the correction. Thank you
Reviewer 3 Report
Comments and Suggestions for Authors
The authors have satisfactorily revised the manuscript.
Comments on the Quality of English Language--
Author Response
Thank you.